# Application of Anti-Inflammatory Agents in Prostate Cancer

**DOI:** 10.3390/jcm9082680

**Published:** 2020-08-18

**Authors:** Koji Hatano, Kazutoshi Fujita, Norio Nonomura

**Affiliations:** 1Department of Urology, Osaka University Graduate School of Medicine, Suita, Osaka 565-0871, Japan; hatano@uro.med.osaka-u.ac.jp (K.H.); nono@uro.med.osaka-u.ac.jp (N.N.); 2Department of Urology, Kindai University Faculty of Medicine, Osakasayama, Osaka 589-8511, Japan

**Keywords:** inflammation, prostate cancer, tumor microenvironment, macrophage, myeloid-derived suppressor cell, chemoprevention, aspirin, metformin, statin, immunotherapy

## Abstract

Chronic inflammation is a major cause of human cancers. The environmental factors, such as microbiome, dietary components, and obesity, provoke chronic inflammation in the prostate, which promotes cancer development and progression. Crosstalk between immune cells and cancer cells enhances the secretion of intercellular signaling molecules, such as cytokines and chemokines, thereby orchestrating the generation of inflammatory microenvironment. Tumor-associated macrophages (TAMs) and myeloid-derived suppressor cells (MDSCs) play pivotal roles in inflammation-associated cancer by inhibiting effective anti-tumor immunity. Anti-inflammatory agents, such as aspirin, metformin, and statins, have potential application in chemoprevention of prostate cancer. Furthermore, pro-inflammatory immunity-targeted therapies may provide novel strategies to treat patients with cancer. Thus, anti-inflammatory agents are expected to suppress the “vicious cycle” created by immune and cancer cells and inhibit cancer progression. This review has explored the immune cells that facilitate prostate cancer development and progression, with particular focus on the application of anti-inflammatory agents for both chemoprevention and therapeutic approach in prostate cancer.

## 1. Introduction

Chronic inflammation plays a major role in the etiology and development of various types of malignant tumors, including hepatocellular carcinoma, gastric cancer, lung cancer, colorectal cancer, and prostate cancer [1,2,3,4,5]. Although inherited germline mutations are involved in prostate cancer development [6,7], immigration studies indicate the importance of environmental factors; for instance, it was found that immigrants from Asian countries in Western countries acquired higher prostate cancer risks within one generation [8,9]. The exposure to environmental factors, such as microbiome, cellular trauma, hormonal imbalances, dietary carcinogens, and obesity, leads to prostate epithelium injury and causes chronic inflammation [3,10,11,12]. In the adult prostate, chronic inflammation is prevalent and associated with putative precursor lesions that can provoke prostate cancer development [13,14,15,16].

A meta-analysis revealed an increased risk of prostate cancer among men with a history of prostatitis, syphilis, and gonorrhea [17]. Although a number of studies supported the idea of a connection between prostatitis and prostate cancer risk [18,19,20,21,22], subsequence studies revealed conflicting results [23,24,25,26,27]. The inconsistent results could be attributed to differences in the study population and potential selection bias, as acute and chronic prostatitis is associated with increased serum prostate-specific antigen (PSA) levels [28]. Previous epidemiological studies focused on the relationship between inflammation and prostate cancer development [29,30,31]. The first prospective study in men without biopsy indication revealed that benign tissue inflammation was positively associated with prostate cancer development [32]. Furthermore, the progression and aggressiveness of prostate cancer was reportedly associated with systemic inflammation markers in the serum, such as C-reactive protein levels, as well as differential blood cell count (neutrophils, lymphocytes, monocytes, and platelets) [33,34,35,36,37].

The pathogenesis of inflammation-associated cancer is complex as both the innate and adaptive immune systems are involved in the process [38,39,40]. Chronic inflammation in the prostate microenvironment causes chronic increase in reactive oxygen species, which is associated with oxidative DNA damage in prostate epithelium [41]. Accumulated DNA damage can cause somatic mutations in key tumor suppressor genes and induce genome instability resulting in genomic changes in oncogenes, thus facilitating the development and progress of prostate cancer [3,42,43]. In fact, in patients with castration-resistant prostate cancer (CRPC), aberrations of *AR* (androgen receptor), E26 transformation-specific genes, *TP53* (tumor protein p53), and *PTEN* (phosphatase and tensin homolog) were frequently observed (40–60% of cases), whereas only few (8%) cases had pathogenic germline alterations [44]. Additionally, inflammatory cells also secrete cytokines and chemokines that stimulate prostate cancer growth, angiogenesis, invasion, and metastasis [45,46,47]. Thus, anti-inflammatory agents are expected to suppress inflammation in the tumor microenvironment and inhibit prostate cancer progression (Figure 1). In this article, we have reviewed the immune cells that facilitate prostate cancer development and progression, and especially focused on the application of anti-inflammatory agents for both chemoprevention and therapeutic approach in prostate cancer.

## 2. Immune Cells Involved in Inflammation and Prostate Cancer Progression

Cancer development and its response to therapy are strongly affected by innate and adaptive immunity, which either promote or attenuate tumorigenesis and can have opposing effects on therapeutic outcome [48]. A large number of studies have focused on immune cells in prostate cancer, including innate immune cells: Macrophages, neutrophils, and mast cells; adaptive immune cells: T cells and B cells; and immune-suppressive cells: regulatory T cells (Treg cells) and myeloid-derived suppressor cells (MDSCs) [49]. In the prostate microenvironment, these immune cells act as either friends or foes [50]. Growing evidence suggests that both macrophages and MDSCs play pivotal roles in inflammation-associated prostate cancer development and progression via down-regulation of effective anti-tumor immunity (Figure 2). Although the precise molecular mechanisms involved are still unclear, animal models of prostate cancer, which mimic the human disease, have contributed to the determination of specific pathways and helped develop novel therapeutic agents.

### 2.1. Macrophages

Tumor-associated macrophages (TAMs) are crucial drivers of tumor promoting inflammation and are generally associated with poor prognosis in solid tumors [51], including prostate cancer [52,53,54,55,56,57,58]. TAMs contribute to tumor progression at different levels by promoting genetic instability, cell proliferation, angiogenesis, and metastasis, as well as suppressing protective adaptive immunity [59,60,61,62,63]. Historically, macrophages are divided into classically activated macrophages (M1) and alternatively activated macrophages (M2), exhibiting anti-tumoral and pro-tumoral properties, respectively. Although interleukin (IL)-4 and IL-13 are the acknowledged signals that regulate M2 polarization of macrophages, recently other subtypes of the M2 class have been identified [64,65,66,67]. Thus, TAM population is referred to as “M2-like” when they include diverse phenotypes that share the functional outputs of tumor promotion and adaptive immunity suppression; these typically express characteristic surface molecules, such as CD163, CD204, and CD206 [62].

Furthermore, elements within the tumor microenvironment, such as hypoxia, fibrosis, cellular stress, and inflammation, dramatically shift the macrophage polarity towards M2-like phenotypes [68,69,70] (Figure 2). Many studies have shed light on the complex signaling network that drives myeloid cells toward M2-like TAMs, which involves various cytokines, chemokines, and signals within the tumor microenvironment [71], including prostaglandin E2 (PGE2) [72], chemokine (C–C motif) ligand (CCL)2 [73], colony stimulating factor (CSF)-1 [74], C–X–C motif chemokine (CXCL)12, and IL-6 [75,76]. Thus, tumor microenvironment can influence TAM polarization by releasing various factors that give rise to a large spectrum of pro-tumoral TAMs [77]. TAMs inhibit effector T cells by secreting IL-10, transforming growth factor (TGF)-*β*, and arginase (ARG)1 as well as via direct cell to cell contact. They also induce Treg cells via IL-10 and TGF-*β* [78,79] (Figure 2). Therefore, TAMs can be potential therapeutic targets, and re-education of pro-tumoral M2-like TAMs toward anti-tumorigenic phenotype may be a potent strategy to treat prostate cancer [80,81].

### 2.2. MDSCs

MDSCs are a tolerogenic and immune-suppressive population of myeloid cells that are significantly expanded in patients with various types of cancers. High MDSC number negatively correlates with disease progression and overall survival, thus suggesting these to be a possible target for cancer immunotherapy [82,83]. MDSCs have distinct phenotypic surface markers as well as functional characteristics, particularly T cell activity inhibition. MDSCs were originally identified in mice as Gr-1^+^ CD11b^+^ cells [84,85]. The Gr-1 marker is not a singular molecule, but a combination of Ly6C and Ly6G markers. Currently, MDSCs include two major subsets based on their phenotypic and morphological features: Polymorphonuclear (PMN)-MDSCs, including CD11b^+^ Ly6C^lo^ Ly6G^+^ cells, and monocytic (M)-MDSCs, including CD11b^+^ Ly6C^hi^ Ly6G^−^ cells. In humans, PMN-MDSC equivalent cells are defined as CD11b^+^ CD14^−^ CD15^+^ or CD11b^+^ CD14^−^ CD66b^+^ and M-MDSCs as CD11b^+^ CD14^+^ HLA-DR^low/−^ CD15^−^ [86]. Growing evidence suggests that MDSCs play an important role in cancer development and progression via suppression of anti-tumoral T cell function in patients with prostate cancer [87,88,89,90,91,92]. However, the MDSC subsets that have clinical relevance during disease progression remain to be identified.

To date, various signaling pathways, including PGE2 [93], CCL2 [94,95], CSF-1 [96], TGF-*β* [97], CXCL5, CXCL12 [98], IL-1*β* [99,100], and IL-6 [101], have been identified to be involved in the infiltration and activation of MDSCs in tumor microenvironment [102,103,104,105,106]. MDSCs suppress anti-tumor immunity through a variety of diverse mechanisms, including ARG1, inducible nitric oxide synthase (iNOS), TGF-*β*, and IL-10, and PMN-MDSCs and M-MDSCs exhibit different mechanisms of immune suppression [107,108,109,110,111,112]. Ultimately, MDSCs inhibit the activation and clonal expansion of tumor-specific T cells, as well as induce Treg cell development (Figure 2). Furthermore, MDSCs secrete various factors that promote prostate cancer progression, such as IL-1 receptor antagonist (IL-1RA) that antagonizes senescence of prostate cancer in a paracrine manner [113], and IL-23 that acts as a driver of CRPC [114]. Thus, targeting MDSCs may provide novel opportunities for cancer therapy.

### 2.3. Crosstalk between Immune Cells, Stromal Cells, and Cancer Cells in Prostate Microenvironment

In the inflammatory prostate microenvironment, crosstalk between immune, stromal, and cancer cells potentially facilitates further tumor progression [115,116,117]. An ex vivo prostate tumor model, derived from patients with prostate cancer, demonstrated that prostate tumors showed low levels of cytotoxic T lymphocytes and T-helper (Th)1 cells-recruiting chemokines, such as CCL5, CXCL9, and CXCL10, but expressed high levels of chemokines implicated in attracting TAMs, MDSCs, and Treg cells, such as CCL2, CCL22, and CXCL12 [118]. CCL22, secreted by tumor cells and TAMs, activates trafficking of Treg cells within the tumor as well as promotes tumor migration and invasion [119,120,121]. Human prostate carcinoma-associated fibroblasts and prostate cancer cells orchestrate and enhance TAM and MDSC recruitment to prostate tumors as well as M2-like TAM differentiation by the chemokines CCL2, CXCL12, and IL-6 during cancer progression [76,122]. Sexually transmitted disease-associated inflammation facilitates IL-6 production by prostate epithelial cells, which induces M2-like TAM polarization [123]. Furthermore, MDSC and tumor cell cross-talk enhances IL-6 production within tumor microenvironment [124], while IL-10, produced by MDSCs, increases M2-like TAMs [125,126,127]. The MDSCs derived from patients with prostate cancer inhibit CD8^+^ T cells through ARG1, a downstream signal transducer and activator of transcription (STAT)3 target gene [91]. Moreover, the phenotypic analysis of prostate infiltrating lymphocytes, derived from patients with prostate cancer, revealed them to be skewed towards a regulatory Treg and Th17 phenotypes [128]. Tregs are associated with poor prognosis and were found to be highly infiltrated in the prostate tissue of patients with prostate cancer [129,130]. Th17 cells, the key mediators in a number of autoimmune diseases, play a role in inflammation-associated prostate cancer [131,132]. Their development depends on the pleiotropic cytokine TGF-*β*, which is also linked to Treg cell development and function [133]. In Hi-Myc mouse model of prostate cancer, retrograde urethral instillation of CP1, a human prostatic isolate of *Escherichia coli,* was reported to induce chronic inflammation characterized by an influx of TAMs and Th17 lymphocytes with distinct cytokine profiles and thus accelerate cancer progression [134].

Crosstalk between MDSCs and mast cells was found to further suppress effective anti-tumor immunity in a transgenic adenocarcinoma of the mouse prostate (TRAMP) model [135]. PhIP (2-amino-1-methyl-6-phenylimidazo[4,5-b]pyridine), one of the most abundant heterocyclic amines in cooked meat, induced rat prostate cancer with elevated DNA mutation frequencies in the prostate as well as infiltration of TAMs and mast cells, suggesting a potential mechanism involving inflammation promotion by which dietary compounds can increase cancer risk [136]. Furthermore, bacterial prostatitis accelerates PhIP-induced prostate carcinogenesis by increasing the level of circulating IL-6 in the rat prostate [137]. Another factor that facilitates chronic inflammation is obesity, wherein adipose derived proinflammatory molecules activate TAMs and MDSCs, subsequently promoting cancer progression [138,139,140,141,142,143,144,145]. CCL2 produced by adipocytes enhances the growth and invasion of prostate cancer cells [146]. Upregulation of serum CCL2 levels enhanced the tumor growth of prostate cancer LNCaP xenografts in high-fat diet fed mice [147]. In obese mice, expanded MDSCs suppress CD8^+^ T cells via iNOS and interferon-*γ*, and also induce M2 TAM polarization via IL-10 [148,149]. The loss of *Pten* in the prostate epithelium causes local MDSC expansion via inflammatory cytokines, such as CSF-1 and IL-1*β*. [96]. In a mouse prostate cancer model driven by loss of *Pten* and *Smad4*, MDSCs play a critical role in cancer progression, as CXCR2-expressing MDSCs infiltrate in the prostate due to CXCL5 up-regulation in tumors [150]. In *Pten*-deficient model mice for prostate cancer, a high-fat diet mediated inflammation-induced M2 TAM differentiation and expansion of MDSCs, accelerated IL-6 secretion, and facilitated tumor growth via IL-6/STAT3 signaling pathway [151]. Thus, the inflammation-associated prostate cancer progression is potentially mediated by diverse mechanisms, such as microbiome, dietary carcinogens, obesity, cellular stress, hypoxia, and fibrosis, which consequently inhibit effective anti-tumor immunity (Figure 2). These pathways may be promising targets for chemoprevention and cancer therapy.

## 3. Chemoprevention of Prostate Cancer

Exposure to various environmental factors can cause chronic inflammation in the prostate and stimulate cancer development and progression. The presence of pathogens in the urogenital organ and gut [152,153] as well as sexual activity [154,155] are associated with chronic inflammation and prostate cancer (Figure 2). The DNA adduct of PhIP, a type of DNA damage, was identified in prostate cancer tissues of patients who frequently consumed cooked red meat [156]. In a cohort of health professionals in the U.S. (*n* = 26,030), intake of PhIP from red meat was significantly associated with prostate cancer risk (HR 1.18, 95% CI 1.03–1.35), especially in high-grade (Gleason score 7 with pattern 4 + 3, and Gleason score 8–10) cancers (HR 1.44, 95% CI 1.04–1.98). [157]. Higher dietary inflammatory index score is associated with a higher risk of incidence and mortality of all cancer types, including prostate cancer [158,159,160]. Consumption of diets rich in anti-inflammatory food components, such as omega-3 fatty acids and green tea, should help prevent prostate cancer [161]. Interestingly, in a mouse prostate cancer MycCaP model, dietary omega-3 fatty acids decreased M2 TAM polarization and inhibited prostate cancer progression [162,163]. Obesity is associated with chronic inflammation and mortality due to various types of malignancies, including prostate cancer [164]. A high-fat diet induced lipid accumulation in prostate tumors, which led to metastatic progression in a nonmetastatic *Pten*-null mouse model [165]. Moreover, epidemiological evidence suggests that metabolic disease, food, and dietary factors are associated with the risk of prostate cancer [166,167,168]. Thus, lifestyle changes are beneficial for prevention of prostate cancer. In addition, anti-inflammatory agents, such as aspirin, non-steroidal anti-inflammatory drugs (NSAIDs), metformin, and statins, may be useful for chemoprevention of prostate cancer (Table 1).

### 3.1. Aspirin and NSAIDs

Aspirin (acetylsalicylic acid), an NSAID, was developed in the late 19th century. As an anti-inflammatory and anti-thrombotic agent, aspirin is widely administrated in order to prevent and treat cardiovascular diseases. Over the last several decades, a number of epidemiological studies demonstrated the association between aspirin use and decreased risk of human carcinogenesis, especially in inflammation-related cancers such as colorectal cancer, gastric cancer, liver cancer, and prostate cancer [169,170,188].

In the Physicians’ Health Study, where healthy male physicians (*n* = 22,071) were randomized to aspirin, *β*-carotene, both, or placebo, current and past regular aspirin use was associated with a lower risk of lethal prostate cancer (current: HR 0.68, 95% CI 0.52–0.89; past: HR 0.54, 95% CI 0.40–0.74) compared to never users [170]. Similarly, in the REDUCE Study, where all men (*n* = 6729) were biopsied independent of PSA levels and had a negative baseline biopsy, the use of aspirin and/or other NSAIDs was significantly associated with decreased total (OR = 0.87, 95% CI 0.76–0.99, *p* = 0.03) and high-grade (OR = 0.80, 95% CI 0.64–0.99, *p* = 0.04) prostate cancer risk [171]. In the Health Professionals Follow-Up Study cohort (*n* = 51,529), long-term higher dose of aspirin (≥ 6 tablets/week) was associated with the reduction of high-grade (HR 0.72, 95% CI 0.54–0.96) and lethal (HR 0.71, 95% CI 0.50–1.00) prostate cancer [172]. In PLCO study (*n* = 29,450) intake of at least one aspirin/day was associated with a lower risk of prostate cancer (HR 0.92, 95% CI 0.85–0.99) compared with never use [173]. In a Finnish Prostate Cancer Screening Trial, an elevated risk of prostate cancer was reported in current NSAID users, but not in previous users, due to protopathic bias as NSAIDs were used to treat symptoms of undiagnosed metastatic prostate cancer as mentioned by the authors [189]. Thus, accumulating evidence indicates the potential inhibitory effect of aspirin on prostate cancer incidence and progression.

Furthermore, in the Physicians’ Health Study, post-diagnostic aspirin use was associated with lower risk of lethal prostate cancer (HR 0.68, 95% CI 0.52–0.90) and overall mortality (HR 0.72, 95% CI 0.61–0.9) [170]. According to the Cancer of the Prostate Strategic Urologic Research Endeavor (CaPSURE) database, aspirin use was associated with a lower risk of prostate cancer-specific mortality (adjusted HR 0.43, 95% CI 0.21–0.87, *p* = 0.02) among patients (*n* = 5955) with localized prostate cancer treated with radical prostatectomy or radiotherapy [174]. Similarly, in the Cancer Prevention Study-II Nutrition Cohort, among men diagnosed with high-risk cancers (≥ T3 and/or Gleason score ≥ 8), post-diagnosis daily aspirin use was also associated with lower prostate cancer-specific mortality (HR 0.60, 95% CI 0.37–0.97) [175]. In a Nationwide Cohort Study (*n* = 29,136), the post-diagnostic use of low-dose aspirin (75–150 mg) slightly reduced prostate cancer mortality over 5-year (HR 0.91, 95% CI 0.83–1.01) and 7.5-year (HR 0.84, 95% CI 0.72–0.97) post-diagnosis exposure periods [176]. An ongoing randomized Add-Aspirin trial (NCT02804815), that began in 2015, aims to determine if post-diagnosis aspirin use improves cancer outcome among men with non-metastatic prostate cancer as well as other types of cancers [190].

Aspirin inhibits cyclooxygenases, COX1 and COX2, which catalyze the production of prostaglandins that trigger pain, fever, blood clotting, or inflammation. Aspirin controls the activity of platelets via COX1 inhibition, which blocks the interaction between platelets and cancer cells [191,192]. In addition, COX2 is frequently expressed in various types of cancers and plays multiple roles in inflammation and cancer progression [193,194]. The histopathological analysis of prostatic tissue derived from patients with prostate cancer revealed that COX2 expression was associated with local chronic inflammation [195]. In a TRAMP model, a COX2 selective inhibitor, celecoxib, inhibited prostate tumor growth and improved survival [196]. COX2 inhibitors suppress prostate cancer growth via a variety of pathways [197,198,199,200,201], including androgen receptor signaling pathway [202,203]. In various types of tumor microenvironment, COX2-mediated PGE2 inhibition suppresses MDSC expansion and provokes a shift in the tumor inflammatory profiles toward anti-cancer immune pathways [204,205,206]. In the prostate tumor microenvironment, PGE2 induces CXCL12 expression in prostate stromal cells [207]. Further, in an ex vivo prostate tumor model, celecoxib suppressed intra-tumoral production of the Treg/MDSC-attractant CXCL12 and Treg-attractant CCL22, while increasing the production of the cytotoxic T lymphocyte (CTL) attractant CXCL10. These changes in local chemokine production were accompanied by the reduced ability of celecoxib-treated tumors to attract Treg cells, and strongly enhanced the attraction of CTLs [118]. As COX2 inhibition also blocks M2 TAM differentiation [208,209], celecoxib suppressed M2 TAMs and local MDSCs, and subsequently inhibited high-fat diet-mediate inflammation, resulting in tumor regression in *Pten*-deficient model mice for prostate cancer [151]. Thus, NSAIDs potentially modulate the immune system, enhance anti-cancer immunity, and inhibit prostate cancer progression.

Long-term administration of COX-2 selective inhibitor is limited for the use of cancer chemoprevention because of the potential cardiovascular risks involved, particularly in older men who are likely to have cardiovascular comorbidities [177,178]. Moreover, neoadjuvant use of short-term celecoxib for 4–6 weeks did not greatly influence COX-related biologic markers in patients with localized prostate cancer who underwent radical prostatectomy [210,211,212]. In the STAMPEDE trial as well, celecoxib daily use for up to one year was insufficient to improve overall survival in patients with advanced hormone naïve prostate cancer who underwent androgen deprivation therapy (ADT), although the preplanned subgroup analyses revealed that celecoxib in combination with zoledronic acid improved overall survival in patients with metastatic disease (HR 0.78, 95% CI 0.62–0.98) [179,180]. Thus, future studies are needed to identify feasible COX2 inhibitor candidates for preventing disease progression in patients with prostate cancer.

### 3.2. Metformin

Metabolic syndrome prevalence was reported to be 55% in men who underwent long-term ADT compared to 22% in those who did not receive ADT [213]. Abdominal obesity and hyperglycemia are responsible for this higher prevalence, predisposing them to higher cardiovascular risk [214]. Thus, metformin, a first-line medication for type 2 diabetes, is beneficial for patients with advanced prostate cancer as it prevents ADT-induced metabolic syndrome [181] and has potential antineoplastic activity in prostate cancer [215]. A recent meta-analysis, which included 30 cohort studies (*n* = 1,660,795), demonstrated that metformin improved both overall survival (HR 0.72, 95% CI 0.59–0.88) and cancer-specific survival (HR 0.78, 95% CI 0.64–0.94) in patients with prostate cancer compared with those not treated with metformin, although the incidence of prostate cancer was not associated with metformin [182]. Importantly, a dose-dependent inverse association between metformin and serum PSA levels was observed, which potentially affects the indication of prostate biopsy as well as detection of prostate cancer [216]. Metformin exerts its anti-cancer effect directly by acting on the tumor via inhibition of the mitochondrial electron transport chain and consequent activation of adenosine monophosphate-activated protein kinase, as well as indirectly by lowering systemic insulin levels [217]. Metformin is also capable of repressing prostate cancer progression by inhibiting infiltration of TAMs, especially those induced by ADT, by inhibiting the COX2/PGE2 axis as observed in the TRAMP model [218]. It also inhibited prostate cancer growth induced by a high-fat diet via inhibition of MDSCs in *Pten*-deficient model mice [219]. Based on its anti-cancer effects and to help prevent the adverse metabolic effects of long-term ADT, metformin is included in the STAMPEDE (Arm K), a randomized controlled phase 3 trial (NCT00268476). Another ongoing randomized phase 3 study aims to determine if metformin can delay the time to progression in men with low risk prostate cancer who underwent active surveillance (NCT01864096).

### 3.3. Statins

Statins are the most commonly used cholesterol-lowering drugs that act by inhibiting HMG-CoA (3-hydroxy-3-methylglutaryl-coenzyme A) reductase activity and are widely used in prevention of coronary artery disease [183]. Emerging evidence suggests that statins may also reduce the risk of cancers. A meta-analysis including 27 studies (*n* = 1,893,571) indicated that statin use reduced the risk of total (RR 0.93, 95% CI 0.87–0.99) and particularly advanced (RR 0.80, 95% CI 0.70–0.90) prostate cancer [184]. In a cohort from Danish registries (statin use *n* = 18,721, no-statin *n* = 277,204), statin use in patients with cancer reduced death due to any cause (HR 0.85, 95% CI 0.83–0.87) as well as due to cancer (HR 0.85, 95% CI 0.82–0.87) [185]. In another cohort from nationwide Danish registries (*n* = 31,790), post-diagnosis statin use was associated with reduced all-cause mortality (HR 0.81, 95% CI 0.76–0.85) and prostate cancer mortality (HR 0.83, 95% CI 0.77–0.89) [186]. A meta-analysis of 13 studies (*n* = 100,536) also showed that post-diagnostic statin use was correlated with reductions in both all-cause mortality (HR 0.77, 95% CI 0.69–0.87) and prostate cancer-specific mortality (HR 0.64, 95% CI 0.52–0.79) [187]. However, statin intake is inversely related to serum PSA levels, which can influence the indication of prostate biopsy [220,221,222,223]. The antineoplastic effect of statins arises from a number of cholesterol-mediated mechanisms, as cholesterol is a key component of lipid rafts, which facilitate intracellular signaling processes such as epidermal growth factor and IL-6 [224]. Importantly, statins inhibit chemokine production, such as CCL2 and CCL5, and act as anti-inflammation agents [225,226]. Therefore, the beneficial effects of statins on the reduction of cardiovascular events as well as cancer-related mortality are attributed to their anti-inflammatory properties. Further studies are needed to identify the mechanism of statin action on chemoprevention of prostate cancer. A phase 3 clinical trial to evaluate impact of atorvastatin on prostate cancer progression during ADT is ongoing (NCT04026230). Another 2 × 2 factorial randomized phase 3 trial was recently launched to evaluate the benefit of aspirin and atorvastatin on overall survival in patients with CRPC (NCT03819101).

## 4. Future Directions: Direct Targeting of Pro-Inflammatory Immunity

There have been substantial advances in the therapeutics of prostate cancer over the past decade. Understanding the mechanism of tumor-promoting chronic inflammation provides novel therapeutic targets for advanced and refractory prostate cancer. Therapeutic strategies against cancer-associated chronic inflammation include: (1) inhibition of pro-tumoral inflammation, (2) boost of anti-cancer pathways, and (3) reprogramming and/or depleting immune cells [227]. TAMs and MDSCs play a vital role in the development of prostate tumor inflammatory microenvironment, which cooperatively inhibits effective anti-tumoral immunity and induces Treg cells. As a result, prostate tumor microenvironment usually becomes immunologically “cold” (Figure 2). Thus, therapeutic strategies that directly target TAMs [228,229,230,231,232,233] as well as MDSCs [109,234,235,236,237,238,239,240] are promising for improved cancer outcomes. Several clinical trials that potentially target TAMs and/or MDSCs have been designed and launched to treat patients with prostate cancer (Table 2).

The IL-6/Janus kinase (JAK)/STAT3 pathway is aberrantly hyperactivated in many types of cancer, including prostate cancer, and such hyperactivation is generally associated with a poor prognosis [241,242,243]. In addition to direct effects on tumor cells, IL-6/JAK/STAT3 signaling has a fundamental effect on tumor-infiltrating immune cells [244,245]. STAT3 negatively regulates dendritic cells, effector T cells, natural killer cells, and neutrophils, suggesting that STAT3 activation in immune cells likely leads to the down-regulation of anti-tumor immunity [246]. At the same time, STAT3 positively regulates TAMs, MDSCs, Treg cells, and Th17 cells [247,248]. Thus, collectively, IL-6/JAK/STAT3 pathway contributes to the development of a highly immunosuppressive tumor microenvironment. In *Pten*-deficient murine model for prostate cancer, inhibition of IL-6 suppressed high-fat diet-mediated prostatic inflammation and subsequent cancer progression [151]. Siltuximab (CNTO 328), an anti-IL-6 monoclonal antibody, was approved by the FDA for treatment in patients with multicentric Castleman’s disease [249]. Inhibition of IL-6 with siltuximab suppressed castration-resistant progression in androgen-dependent prostate cancer xenograft model mice [250]. However, phase 2 clinical trials (NCT00433446, NCT00385827) revealed that siltuximab had no significant clinical benefit in patients with metastatic CRPC who showed a dramatic increase in plasma IL-6 after treatment and confirmed the poor prognosis associated with elevated IL-6 at baseline [251,252]. As analyses of radical prostatectomy specimens from a phase 1 study revealed a decrease in phosphorylated STAT3 and mitogen-activated protein kinases by siltuximab treatment, IL-6 blockade may be effective early during disease progression [253]. Another FDA-approved drug called niclosamide, an oral antihelminthic drug, exerts its effects via Wnt/*β*-catenin, mammalian target of rapamycin complex 1, STAT3, NF-κB, and Notch signaling pathways. Therefore, it potentially has broad clinical applications for the treatment of diseases other than those caused by parasites, including metabolic diseases, infection, and cancer [254]. Niclosamide suppresses macrophage-induced inflammation via STAT3 and/or NF-κB signaling [255]. In addition, niclosamide was identified as a potent androgen receptor splice variant 7 (AR-V7) inhibitor in prostate cancer cells [256]. Clinical trials to evaluate impact of niclosamide combined with enzalutamide or abiraterone in patients with metastatic CRPC are ongoing (NCT03123978, NCT02807805). Antisense oligonucleotides provide another distinctly different approach to inhibit cellular STAT3, inhibiting immunosuppressive MDSCs and enhancing anti-tumor immunity, which could effectively eradicate prostate tumors in a mouse model [257].

CSF-1 is a major survival factor for TAMs, as CSF-1 receptor inhibition strongly reduces TAMs and increases cytotoxic CD8^+^ T cells in animal models [258]. CSF-1 is reportedly increased in irradiated prostate tumors, which enhances tumor-infiltrating TAMs and MDSCs that can limit the efficacy of radiotherapy in prostate tumor murine model [259]. ADT-induced prostate cancer cells express CSF-1 and other cytokines that significantly increase M2 TAM infiltration and potentially cause castration-resistant cancer progression [260]. Pexidartinib (PLX3397), an FDA-approved CSF-1 receptor inhibitor, shows a robust tumor response in tenosynovial giant cell tumor [261]. The addition of pexidartinib to docetaxel improved therapeutic efficacy in CRPC by reducing the pro-tumorigenic influences of TAMs in mouse xenograft models [262]. A clinical trial to evaluate the impact of pexidartinib combined with radiation therapy and ADT in patients with localized prostate cancer is ongoing (NCT02472275).

CCL2, also known as monocyte chemoattractant protein (MCP)-1, is a potent monocyte-attracting chemokine and greatly contributes to the recruitment of peripheral blood monocytes into sites of inflammatory responses and tumors [263,264]. In prostate cancer, CCL2 is particularly up-regulated in bone metastasis, and promotes tumor growth and progression [265,266,267,268,269]. Systemic administration of CCL2 monoclonal antibody, carlumab (CNTO 888), in VCaP xenograft model mice attenuated TAM infiltration and retarded tumor growth. Thus, CCL2 acts as a mediator of prostate cancer growth through the regulation of TAMs [270]. Moreover, inhibition of CCL2 by carlumab combined with docetaxel significantly reduced tumor burden compared with docetaxel alone in prostate cancer xenograft model mice [271]. However, carlumab could not effectively block serum CCL2 levels and showed no anti-tumor activity as a single agent in metastatic CRPC in the clinical trial (NCT00992186) [272].

CXCL12, also known as stromal cell-derived factor (SDF)-1, is a strong chemotactic for lymphocytes as well as myeloid cells, including TAMs and MDSCs [273,274,275]. CXCL12 and its receptor, CXCR4, play pivotal roles in tumor development, progression, angiogenesis, and metastasis in various types of cancers [276]. The expression of CXCR4 protein is significantly associated with the presence of bone metastasis in prostate cancer [277]. Inhibition of CXCR4 by plerixafor (AMD3100) or CTE9908 significantly reduced bone metastasis in prostate cancer model mice [278], as the tumorigenic potential is largely regulated by the CXCR4 signaling pathway in prostate cancer cells [279]. Obesity causes inflammation-mediated tumor progression via CXCL12–CXCR4/CXCR7 signaling axis which was attenuated by plerixafor in *Myc*-induced prostate tumor model mice [280]. Plerixafor inhibits tumor–stroma interactions through CXCL12/CXCR4 pathway, enhancing efficacy of docetaxel in prostate cancer [281]. Although plerixafor is approved by the FDA for autologous transplantation in patients with Non-Hodgkin’s lymphoma or multiple myeloma, it can also be used in various other malignancies and immunological disorders [282].

Tasquinimod (ABR-215050), an oral immunomodulatory compound, reportedly affects the accumulation and function of tumor-suppressive myeloid cells, MDSCs, and M2-like TAMs via targeting the inflammatory protein S100A9 [283,284,285]. In a randomized phase 3 trial (NCT01234311), chemotherapy-naive men with metastatic CRPC (*n* = 1245) were randomly assigned either tasquinimod or placebo. Tasquinimod significantly improved radiographic progression-free survival compared with placebo (HR 0.64, 95% CI 0.54–0.75), although no overall survival benefit was observed [286]. Bruton tyrosine kinase (BTK) plays a crucial role in B cell development as well as T2 TAM polarization. Inhibition of BTK with an FDA-approved inhibitor, ibrutinib, restores T cell-dependent anti-tumor immune responses, and potentially inhibits the progression of solid tumors. [287,288]. A clinical trial to evaluate the efficacy of ibrutinib on localized prostate cancer is ongoing (NCT02643667). Recently, a substantial number of studies focused on immunometabolism, particularly the changes in intracellular metabolic pathways in immune cells that alter their function [289]. Among these, tryptophan metabolism catalyzed by the enzyme indoleamine 2,3-dioxygenase (IDO) was reported to be crucial for anti-tumor immunity [290]. IDO is widely overexpressed in various types of cancers and inhibits CD8^+^ effector T cells and natural killer cells, while activating Treg cells and MDSCs [291,292]. Accordingly, inhibitors of the enzymatic activity and effector functions of IDO have been developed as tools for novel cancer therapy [293,294]. Indoximod, an IDO1 pathway inhibitor, was used to treat patients with metastatic CRPC in combination with Sipuleucel-T in a phase 2 clinical trial (NCT01560923).

Immune checkpoint inhibitors which target cytotoxic T lymphocyte-associated antigen 4 (CTLA-4) and the programmed cell death protein 1 pathway (PD-1/PD-L1) have presented substantial benefits for many types of cancers, but only a marginal benefit for prostate cancer, because of the immunosuppressive tumor microenvironment and low mutation burden [295,296]. However, novel strategies are emerging to modulate the prostate immune-suppressive microenvironment. The preclinical studies demonstrated that the combination therapy with immune checkpoint inhibitors and agents targeted at pro-inflammatory immunity could effectively eradicate prostate cancer [297,298,299,300]. Thus, recent advances may shed new light on immunotherapy for prostate cancer.

## 5. Conclusions

Chronic inflammation plays a major role in the etiology, development, and progression of prostate cancer. A number of studies demonstrated that aspirin is a potent chemopreventive agent for prostate cancer, concurrently preventing cardiovascular diseases. Metformin and statins may also have potential benefit for chemoprevention of prostate cancer. Crosstalk between immune cells and cancer cells orchestrates the generation of prostatic inflammatory microenvironment via a variety of cytokines and chemokines. Importantly, TAMs and MDSCs are significantly expanded in patients with prostate cancer, causing effector T cell inhibition and Treg cell induction, thus providing possible targets for cancer immunotherapy. Recent years have seen emergence of novel strategies that target pro-inflammatory immunity to treat patients with prostate cancer.

## Figures and Tables

**Figure 1 jcm-09-02680-f001:**
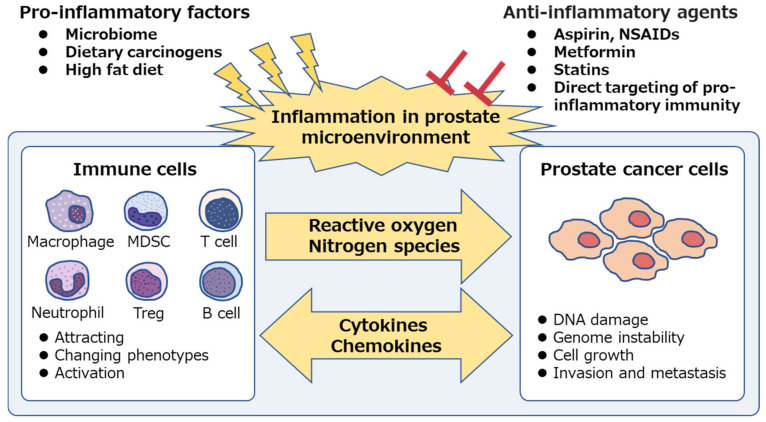
Chronic inflammation is associated with prostate cancer. Pro-inflammatory factors, such as microbiome and dietary components, are the potential cause of prostatic inflammation. Immune cells secrete reactive oxygen and nitrogen species, induce DNA damage and genome instability in prostate epithelium and cause prostate cancer development. Both immune cells and prostate cancer cells secrete intercellular signaling molecules, such as cytokines and chemokines, and contribute to the generation of inflammatory microenvironment, which facilitates cancer progression. Anti-inflammatory agents suppress the “vicious cycle” and inhibit prostate cancer development and progression. NSAIDs, non-steroidal anti-inflammatory drugs; MDSC, myeloid-derived suppressor cell; Treg, regulatory T cell.

**Figure 2 jcm-09-02680-f002:**
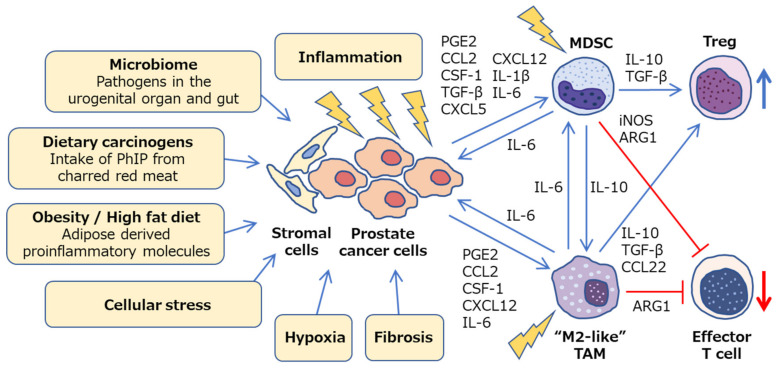
Inflammatory microenvironment in prostate cancer. Diverse mechanisms, such as microbiome, dietary carcinogens, obesity, cellular stress, hypoxia and fibrosis, can be a potential cause of inflammation in prostate cancer. Crosstalk between cancer cells, stromal cells, and immune cells promotes chronic inflammation and facilitates prostate cancer progression via a variety of intercellular signaling molecules. Prostate tumor microenvironment is generally considered to be immunologically “cold” as M2-like TAMs and MDSCs cooperatively inhibit effector T cells and activate regulatory T cells. PhIP, 2-amino-1-methyl-6-phenylimidazo [4,5-b]pyridine; PGE2, prostaglandin E2; CSF-1, colony stimulating factor-1; TGF-*β*, transforming growth factor-*β*; ARG1, arginase 1; iNOS, inducible nitric oxide synthase; TAM, tumor-associated macrophage; MDSC, myeloid-derived suppressor cell; Treg, regulatory T cell.

**Table 1 jcm-09-02680-t001:** Overview of potential applications of anti-inflammatory agents in chemoprevention of PCa.

Agents	Summary of Findings	References
Aspirin	1. Aspirin, which is widely administrated in order to prevent and treat cardiovascular diseases, potentially reduces a risk of human carcinogenesis.	[169]
	2. Current and past regular aspirin use is likely to be associated with a lower risk of PCa.	[170,171,172,173]
	3. Post-diagnosis aspirin use is likely to be associated with lower PCa-specific mortality.	[170,174,175,176]
COX-2 inhibitors	1. Long-term use of COX-2 selective inhibitor is associated with the potential cardiovascular risks.	[177,178]
	2. Feasible COX2 inhibitor candidates are still unclear for preventing PCa progression.	[179,180]
Metformin	1. Metformin is beneficial for patients with PCa, preventing ADT-induced metabolic syndrome.	[181]
	2. Metformin may improve PCa-specific survival, although the incidence of PCa is not associated with metformin.	[182]
Statins	1. Statins are widely used in the prevention of coronary artery disease.	[183]
	2. Statin use may reduce the risk of PCa.	[184]
	3. Post-diagnostic statin use may correlate with reductions in PCa-specific mortality.	[185,186,187]

PCa, prostate cancer; ADT, androgen deprivation therapy.

**Table 2 jcm-09-02680-t002:** Clinical trials of agents targeted at TAMs and/or MDSCs in prostate cancer.

Drug Name	Target	Inhibitor Type	Phase	Indication	Combination	ClinicalTrials.gov Reference
CNTO328 (siltuximab)	IL-6	mAb	1	mCRPC	DOC	NCT00401765 ^b^
CNTO328 (siltuximab)	IL-6	mAb	2	mCRPC	NA	NCT00433446 ^b^
CNTO328 (siltuximab)	IL-6	mAb	2	mCRPC	MIT + Pred	NCT00385827 ^b^
Ruxolitinib	JAK1/2	SM	2	mCRPC	NA	NCT00638378 ^b^
Niclosamide	STAT3	SM	1	mCRPC	ENZ	NCT02532114 ^b^
Niclosamide	STAT3	SM	1	mCRPC	ENZ	NCT03123978 ^a^
Niclosamide	STAT3	SM	2	mCRPC	ABI + Pred	NCT02807805 ^a^
PLX3397 (Pexidartinib)	CSF1R	SM	1	Intermediate or high risk PCa	RT + ADT	NCT02472275 ^a^
PLX3397 (Pexidartinib)	CSF1R	SM	2	mCRPC	NA	NCT01499043 ^b^
JNJ-40346527	CSF1R	SM	1	High risk localized PCa	RP	NCT03177460 ^a^
LY3022855 (IMC-CS4)	CSF1R	mAb	1	Advanced PCa and BCa	NA	NCT02265536 ^b^
CNTO 888 (Carlumab)	CCL2	mAb	2	mCRPC	NA	NCT00992186 ^b^
Burixafor hydrobromide	CXCR4	SM	1	mCRPC	± G-CSF ± DOC	NCT02478125 ^b^
Tasquinimod	S100A9	SM	3	mCRPC	NA	NCT01234311 ^b^
Ibrutinib	BTK	SM	1, 2	Localized PCa	RP	NCT02643667 ^a^
Indoximod	IDO	SM	2	mCRPC	Sipuleucel-T	NCT01560923 ^b^

mAb, monoclonal antibody; SM, small molecule; mCRPC, metastatic castration resistant prostate cancer; PCa, prostate cancer; BCa, breast cancer; DOC, docetaxel; NA, not applicable; MIT, mitoxantrone; Pred, Prednisone; ENZ, enzalutamide; ABI, Abiraterone; RT, radiation therapy; ADT, androgen deprivation therapy; RP, Radical Prostatectomy. ^a^ Trial currently ongoing. ^b^ Trial completed.

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
