# Peer review of "Application of Anti-Inflammatory Agents in Prostate Cancer"

_jcm, 2020, doi:10.3390/jcm9082680_

Round 1

Reviewer 1 Report

The authors have written a thorough review on the topic of inflammation in prostate cancer.  However, perhaps because of its thoroughness, and because there are few illustrations, physicians in clinical practice will likely have great difficulty reading through it, or even scanning it to determine its major points.

Here are my suggestions:

Since metformin doesn't seem to reduce incidence of prostate cancer, but aspirin and statins may, reword the sentence about them in the abstract to "potential application in chemoprevention and treatment of prostate cancer."

Please expand Figure 2 to a whole page, and include the factors influencing the immune response included in the text, other than inflammation: hypoxia, fibrosis, cellular stress, high-fat diet, diet high in red meat, obesity, and infection.

Subsequently, when you discuss these risk factors, consider placing an insert showing a portion of Figure 2 which relates to that risk factor.  Alternately, you could bold a summary sentence at the end of the paragraph about how that risk factor acts.  

The other new component of the expanded Figure 2 should be treatments which push the immune system into tumor suppression, rather than tumor-enhancement.  You could show figure inserts or summary bolded sentences in the same way as above.

Since large studies showing real benefit are best seen with aspirin, and to a lesser extent COX-2 inhibitors, metformin, and statins, I would acknowledge that fact in the conclusion, and perhaps in the abstract.

Without tracking down the references on "Future Directions", it is hard to know how solid the research is on the various interventions described.  Yet you spend 3 full pages on them, out of nearly 11 pages total, and you also mention the checkpoint inhibitors in the abstract--is this warranted?  If there is very little evidence, I would shorten, and de-emphasize this section.

Thank you.

Reviewer 2 Report

Paper Title: Application of Anti-inflammatory Agents in Prostate Cancer.

General Comments:

There is no potential bias in reviewing this manuscript. This manuscript has been written with a clear objective in mind. Overall, this review pays particular attention to the application of anti-inflammatory agents for both chemoprevention and therapeutic approach in prostate cancer. It introduces the topic of discussing chronic inflammation and then goes to describe the different immune cells involved in prostate cancer progression. It then summaries the different chemoprotective agents and discusses the future direction of possible targeting of pro-inflammatory immunity. The authors have adequately reviewed the literature on the different chemoprotective agents and discusses about future direction.

There is a lot of information to follow and maybe a more structured approach would align most of the information presented. For example, the use of a table format to showcase the different chemopreventative agents. The discussion and conclusion can be further work on with examples and explanation on the findings with relevant information and points to substantiate the claims.

The Abstract

The abstract is clear and concise detail all the important aspect of the review.

The Introduction

The introduction is concise and relevant to the proposed findings. The main purpose of the study is clearly defined and explained. The authors did provide a rationale for performing the study and the reasons described are substantial.

The chemoprevention of prostate cancer section

This review section has been written with much thought however, as mentioned a table listed each of the different trails on each medication would assist.

The Discussion Section

The discussion does have enough critical review of the literature to substantiate the points the authors made. However, it can be further substantiated with a more critical comparison should be made to the current literature on why the authors' suggestion can provide potential pathways and further enhance the knowledge.
